# Experimental study on mechanical response and crack evolution law of coal and sandstone under different stress environments

**Wenbao Shi, Qingzhao Xu** ⓘ *, **Zhuang Miao, Chao Qi, Jucai Chang, Chuanming Li, Aoyun Yan**

School of Mining Engineering, Anhui University of Science and Technology, Huainan, Anhui, China

\* xqz990523@163.com

**Data Availability Statement:** All relevant data are within the manuscript and its Supporting Information files.

## Abstract

In order to study the mechanical response and crack evolution law of different lithologic rock bodies under different stress environments in deep stress mines, based on the deviator stress theory, the actual triaxial disturbance unloading rock test system was used to simulate the stress occurrence environment of the original rock. The mechanical characteristics of different $\sigma_2$ coal rock masses were studied, and the crack evolution law of coal and sandstone under different stress environments was analyzed. The results show that the increase of $\sigma_2$ inhibits the deformation in the $\sigma_2$ direction of coal and sandstone, promotes the expansion and deformation in the $\sigma_3$ direction, and enhances its peak strength and elastic modulus. The development characteristics of internal cracks in rock mass are directly related to the stress environment, and the increase of $\sigma_2$ promotes the increase of the proportion of coal $RA$ value, weakens the proportion of sandstone $RA$ value, aggravates the development of coal internal shear cracks, and inhibits the development of internal shear cracks in sandstone. The larger $\sigma_2$, the greater the initial AE ringing count of coal and sandstone, and the greater the AE cumulative energy when the rock mass is finally damaged. At the same time, due to the self-organizing behavior in the process of crystal failure in sandstone, the cumulative energy curve of sandstone fluctuates in a step-like manner. The ringing count and cumulative energy increase suddenly, which can predict the imminent instability and failure of the rock, and the research results can provide an experimental basis for the sudden instability of deep high-stress roadways.

## Introduction

In the deep environment, the coal and rock mass stress environment presents the complex stress characteristics of "three highs and one disturbance" [1]. Due to the influence of excavation activities, the overlying strata of the roadway have a large range of movement and stress redistribution, resulting in stress concentration, especially the bearing stress change of the rock mass in front of the working face [2]. The rock mass in front of the working face is affected by mining, and the stress environment undergoes the stress environment

**Funding:** This work was supported by the National Natural Science Foundation of China (Grant Nos.52104117-SWB and 52174105-CJC and 52174103-LCM) and Excellent Scientific Research and Innovation Team (2023AH010023).

**Competing interests:** The authors declare no competing interests.

transformation process of original rock stress, stress increase, and stress decrease. It is not difficult to find that roadway roofs' collapse and fracture accidents are closely related to the change of stress field around the surrounding rock. Therefore, it is essential to study the mechanical response characteristics and deformation and failure law of rock mass under different stress environments to ensure the overall stability of the roadway surrounding rock.

In terms of rock mass stress field, Zhang et al. [3]. Through the conventional triaxial tests of sandstone, it is concluded that with the increase of confining pressure, the rock's elastic modulus, peak strength, and fracture closure stress are positively correlated with the confining pressure. At the same time, Poisson's ratio shows the change characteristics of increasing first and then decreasing, which weakens the degree of damage to the rock. Chen et al. [4] carried out different confining pressure experiments on the jointed rock mass through the bond model and found that the rock mass's compressive strength and elastic modulus were positively correlated with the confining pressure. The increase in confining pressure inhibited the crack development and propagation of the rock mass, resulting in the change of its peak failure morphology. Zhang et al. [5] found that with the increase of confining pressure, the failure mode changed from single splitting failure to multi-crack dendritic failure, and the fracture surface was powdery near the fracture surface under high confining pressure. Wu et al. [6] found that the increase in confining pressure can effectively reduce the microscopic roughness of the rock. However, these studies are carried out under the condition of equal confining pressure, and the real stress state of underground rock mass is three-way unequal pressure [7–9], so there are some deficiencies in analyzing the mechanical characteristics and failure mode of rock mass under equal confining pressure.

The development and application of a true triaxial testing machine [10, 11] makes up for the defects of conventional triaxial. It can independently apply different principal stresses to each surface of rock mass. Based on this, many researchers simulate the complex stress environment of deep rock mass by changing the loading path. However, due to the influence of mining activities, the original stress field of the rock mass changes, and the rock mass appears to be a single-sided unloading phenomenon. Many researchers have simulated the influence of the intermediate principal stress through the actual triaxial test machine for this phenomenon and concluded that the intermediate principal stress could significantly affect the rock strength and failure characteristics [12–14]. However, the damage and micro-crack development caused by rock mass during stress loading are invisible. There are many methods for monitoring rock mass damage and crack development [15, 16]. As a non-destructive monitoring method, acoustic emission (AE) can identify and calculate damage through the sound waves transmitted during loading. Therefore, it is widely used in rock mass damage [17–19]. At present, for the study of acoustic emission, Niu et al. [20] found that the spatial distribution and propagation trend of the number of AE crack events in the rock mass is consistent with the macroscopic crack propagation results on the surface of the specimen through uniaxial compression experiments on coal and sandstone. Su et al. [21] used the accurate triaxial loading system to analyze the acoustic emission characteristics of granite in rockburst. The occurrence of a large number of low, medium, and high dominant frequency information of acoustic emission in a period before the quiet period was used as the precursor information of rockburst. Many scholars mostly carry out different stress path loading and unloading tests for single lithology rock mass and analyze rock's deformation and failure characteristics through acoustic emission characteristics. The research results provide sound guidance and reference for this paper. However, in the process of roadway excavation, the surrounding rock mass is not a single lithology, and the phenomenon of composite rock mass often accompanies it. Compared with single lithology, the surrounding rock of composite rock mass roadway has more complexity.

Therefore, in this paper, the actual triaxial loading test is carried out on coal and sandstone through the actual triaxial disturbance unloading rock test system and the acoustic emission test system. The mechanical response characteristics of different lithological rocks under the same stress environment are analyzed. The crack development mechanism of different lithologies under different stress environments is revealed by combining the acoustic emission system, which provides a basis for the sudden instability of the surrounding rock of the roadway of the deep composite rock mass.

## The deviatoric stress field theory of surrounding rock of intermediate principal stress roadway

After the rock mass excavation, due to roadway and development, the stress redistribution of surrounding rock occurs, and the orientation and size of principal stress change accordingly. The stress distribution is shown in Fig 1.

In the figure, $\sigma_z$ and $\sigma_x$ are the vertical stress and horizontal stress of the vertical borehole axis, respectively. $\sigma_v$ is the principal stress parallel to the borehole axis; $\sigma_r$, $\sigma_\theta$, $\tau_{r\theta}$, and $\sigma_z$ are the radial stress, tangential stress, shear stress, and the principal stress perpendicular to the axial direction of the borehole. From the generalized plane strain theory, the radial, tangential, and shear stress of the surrounding rock of the roadway and borehole can be obtained. The relationship between the far-field stress of the surrounding rock and the radius of the borehole is as shown in Formula (1) [22, 23]:

$$\begin{cases} \sigma_r = \dfrac{1}{2}(\sigma_x + \sigma_z)\left(1 - \dfrac{R^2}{r^2}\right) - \dfrac{1}{2}(\sigma_z - \sigma_x) \times \left(1 - 4\dfrac{R^2}{r^2} + 3\dfrac{R^4}{r^4}\right)\cos(2\theta) \\[2mm] \sigma_\theta = \dfrac{1}{2}(\sigma_x + \sigma_z)\left(1 + \dfrac{R^2}{r^2}\right) + \dfrac{1}{2}(\sigma_z - \sigma_x) \times \left(1 + 3\dfrac{R^4}{r^4}\right)\cos(2\theta) \\[2mm] \tau_{r\theta} = \dfrac{1}{2}(\sigma_x - \sigma_z)\left(1 + 2\dfrac{R^2}{r^2} - 3\dfrac{R^4}{r^4}\right)\sin(2\theta) \\[2mm] \sigma_v = \sigma_y - 2\mu(\sigma_x - \sigma_z)\dfrac{R^2}{r^2}\cos(2\theta) \end{cases} \quad (1)$$

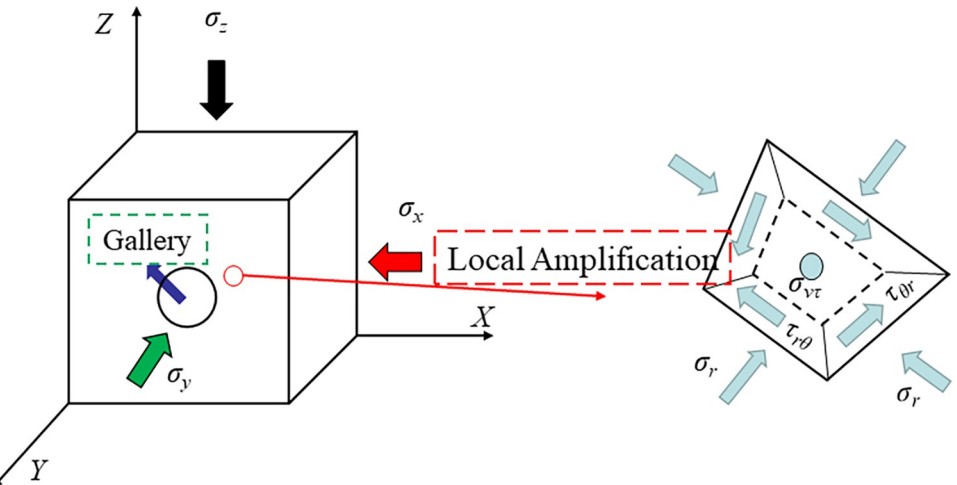

**Fig 1. Stress distribution of roadway surrounding rock.**

In the formula, $R$ is the radius of the roadway; $r$ is the distance between the surrounding rock unit and the center of the roadway; $\theta$ is the angle between the position of the surrounding rock unit and the horizontal direction; $\mu$ is the Poisson's ratio of surrounding rock.

On the plane perpendicular to the Y axis in Fig 1, the principal stress transformation formula of the roadway surrounding the rock unit is:

$$\begin{cases} \sigma_1^* = \dfrac{\sigma_r + \sigma_\theta}{2} + \sqrt{\left(\dfrac{\sigma_r - \sigma_\theta}{2}\right)^2 + \tau_{r\theta}^2} \\ \sigma_2^* = \sigma_v \\ \sigma_3^* = \dfrac{\sigma_r + \sigma_\theta}{2} - \sqrt{\left(\dfrac{\sigma_r - \sigma_\theta}{2}\right)^2 + \tau_{r\theta}^2} \end{cases} \tag{2}$$

In the formula: $\sigma_1^*$, $\sigma_2^*$, $\sigma_3^*$ is the principal stress of the surrounding rock unit of the roadway.

Because the roadway is in a three-dimensional stress state, the magnitude of $\sigma_x$, $\sigma_y$, $\sigma_z$ stress is unknown, and it is impossible to judge the magnitude of the principal stress $\sigma_1^*$, $\sigma_2^*$, $\sigma_3^*$. Therefore, it needs to be classified and discussed. It can be clearly seen from Formula (2). that no matter what stress state the roadway is in, there is $\sigma_1^* \geq \sigma_3^*$, and the order of magnitude of $\sigma_2^*$ and $\sigma_1^*$, $\sigma_3^*$ has three cases: (1) $\sigma_1^* \geq \sigma_3^* \geq \sigma_2^*$; (2) $\sigma_1^* \geq \sigma_2^* \geq \sigma_3^*$; (3) $\sigma_2^* \geq \sigma_1^* \geq \sigma_3^*$. In this paper, only the second stress environment analysis is selected.

The calculation formula of the maximum and minimum principal deviatoric stress $S_1$ and $S_3$ is (3):

$$\begin{cases} S_1 = \sigma_1 - \sigma_m \\ S_3 = \sigma_3 - \sigma_m \\ \sigma_m = (\sigma_1 + \sigma_2 + \sigma_3)/3 \end{cases} \tag{3}$$

The formula (2) is brought into the formula (1) to calculate the principal stress value of the roadway, and then the principal stress value is brought into the formula (3) to calculate the principal deviatoric stress value formula (4) as follows:

$$\begin{cases} S_1 = \sqrt{\left[\dfrac{R^2}{r^2}\left(\dfrac{\sigma_x + \sigma_z}{2}\right) - \left(\dfrac{\sigma_x - \sigma_z}{2}\right)\left(\dfrac{3R^4}{r^4} - \dfrac{2R^2}{r^2} + 1\right)\cos(2\theta)\right]^2 + \left(\dfrac{\sigma_x - \sigma_z}{2}\right)^2\left(\dfrac{2R^2}{r^2} - \dfrac{3R^4}{r^4} + 1\right)^2\sin(2\theta)} + \\ \left[\dfrac{\sigma_x + \sigma_z}{6} - \dfrac{\sigma_y}{3} - \dfrac{(\sigma_x - \sigma_z)R^2\cos(2\theta)}{3r^2}\right] + \dfrac{2R^2\mu(\sigma_x - \sigma_z)\cos(2\theta)}{3r^2} \\ S_3 = -\sqrt{\left[\dfrac{R^2}{r^2}\left(\dfrac{\sigma_x + \sigma_z}{2}\right) - \left(\dfrac{\sigma_x - \sigma_z}{2}\right)\left(\dfrac{3R^4}{r^4} - \dfrac{2R^2}{r^2} + 1\right)\cos(2\theta)\right]^2 + \left(\dfrac{\sigma_x - \sigma_z}{2}\right)^2\left(\dfrac{2R^2}{r^2} - \dfrac{3R^4}{r^4} + 1\right)^2\sin(2\theta)} + \\ \left[\dfrac{\sigma_x + \sigma_z}{6} - \dfrac{\sigma_y}{3} - \dfrac{(\sigma_x - \sigma_z)R^2\cos(2\theta)}{3r^2}\right] + \dfrac{2R^2\mu(\sigma_x - \sigma_z)\cos(2\theta)}{3r^2} \end{cases} \tag{4}$$

The research on the failure theory and support technology of the surrounding rock of the roadway is mainly based on the stress of the surrounding rock, the distribution characteristics of the stress of the surrounding rock, and the interaction force between the surrounding rock and the supporting body. From the perspective of rock and soil mechanics, the occurrence of plastic deformation is mainly determined by deviant stress, and the surrounding rock stress composed of the superposition of original rock stress and deviator stress will inevitably lead to the instability and failure of the surrounding rock within a specific range under the action of

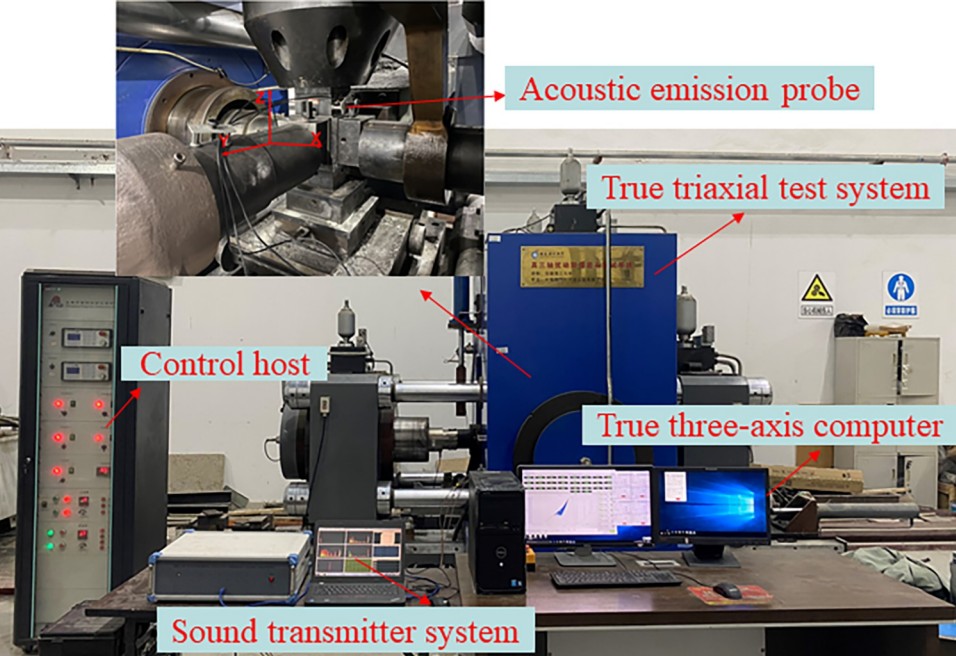

**Fig 2. Experimental instruments.**

deviant stress, and the stress peak will be transferred to the deep part of the roadway. Hence, the influence of deviator stress on rock plastic failure is of great significance.

## Experimental equipment and scheme

### Laboratory equipment

The test system was adopted, composed of a dynamic disturbance system and a three-way loading system, and the cube rock specimens were loaded in three directions and six sides through three independent loading systems. The acoustic emission adopts the Beijing Soft Island DS5 acoustic emission system, with three acoustic emission probes to collect signals. The sampling frequency is 40dB for the preamplifier (gain), and to minimize the impact of noise, the threshold value is set to 50dB, and the acoustic emission sampling frequency range is set to 1kHz~1MHz (Fig 2).

### Experimental scheme

The coal and sandstone required for this test come from the Guqiao Coal Mine of Anhui Huai-nan Mining Group Co., Ltd. The large coal and sandstone body taken on site are processed into a cube shape of 100mm×100mm×100mm (unevenness ≤0.02mm, size error ≤0.2mm) by laboratory means to ensure that the accuracy of the sample meets the requirements, the specific coal and sandstone samples are shown in Fig 3, and the uniaxial compressive strength of the test coal is 21.665MPa, The Poisson's ratio is 0.155, the uniaxial compressive strength of sandstone is 42.608MPa, and the Poisson's ratio is 0.059. The triaxial servo-controlled testing machine preloads the specimen in three directions simultaneously. The preset pressure values in the three directions are reached simultaneously. Then, $\sigma_3$ and $\sigma_2$ remain unchanged, and $\sigma_1$ increases at 30kN/min until the sample is damaged and unstable. According to the schematic diagram of the stress distribution of the surrounding rock of the roadway (Fig 4), it can

**Fig 3. Coal and sandstone specimens.**

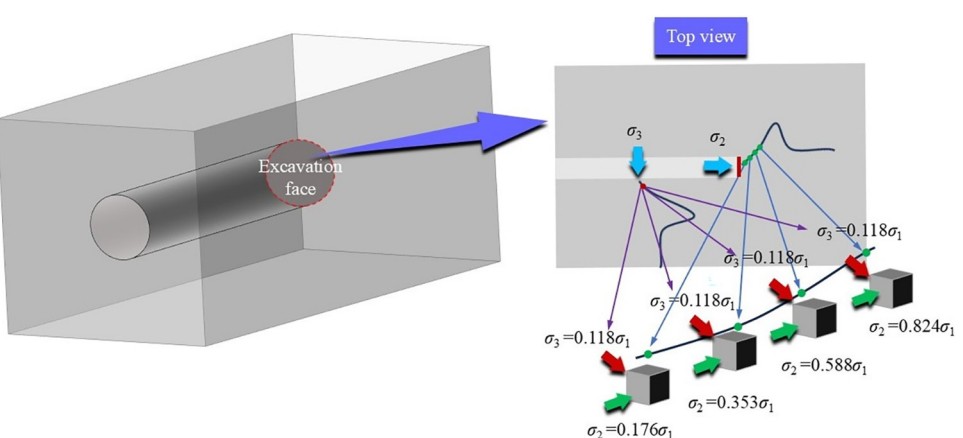

**Fig 4. Schematic diagram of stress distribution in surrounding rock of roadway.**

**Table 1. Loading method.**

| Sample number | Dimensions of test pieces | Default value | Preset value loading speed | The second stage loading method |
|---|---|---|---|---|
| 1-1(Coal, sandstone) | 100×100×100 | $\sigma_1$ = 17MPa | 16.7kN/min | 30kN/min Add to destruction |
|  |  | $\sigma_2$ = 3MPa | 2.8kN/min | Stable invariant |
|  |  | $\sigma_3$ = 2MPa | 1.8kN/min | Stable invariant |
| 1-2(Coal, sandstone) | 100×100×100 | $\sigma_1$ = 17MPa | 16.7kN/min | 30kN/min Add to destruction |
|  |  | $\sigma_2$ = 6MPa | 5.8kN/min | Stable invariant |
|  |  | $\sigma_3$ = 2MPa | 1.8kN/min | Stable invariant |
| 1-3(Coal, sandstone) | 100×100×100 | $\sigma_1$ = 17MPa | 16.7kN/min | 30kN/min Add to destruction |
|  |  | $\sigma_2$ = 10MPa | 9.8kN/min | Stable invariant |
|  |  | $\sigma_3$ = 2MPa | 1.8kN/min | Stable invariant |
| 1-4(Coal, sandstone) | 100×100×100 | $\sigma_1$ = 17MPa | 16.7kN/min | 30kN/min Add to destruction |
|  |  | $\sigma_2$ = 14MPa | 13.8kN/min | Stable invariant |
|  |  | $\sigma_3$ = 2MPa | 1.8kN/min | Stable invariant |

be seen that due to the change of the principal stress in the process of excavation of the roadway, as the rock mass is far away from the free side, the horizontal stress shows a transition from the direction of stress increase—stress reduction—original stress, at this time, the maximum principal stress is the vertical principal stress (considering the buried depth is 680m, the vertical principal stress $\sigma_1$ = 17MPa), the position of the unit body is selected in the roadway gang and near the free side, so the minimum principal stress $\sigma_3$ = 2MPa is set, and at the same time, The rock mass at the roadway excavation face has no supporting effect, which forces the rock mass in front of it to produce slow deformation behavior, resulting in changes in the horizontal stress at different positions of the rock mass in front of the roadway (intermediate principal stress $\sigma_2$). Therefore, to study the influence of different intermediate principal stresses on the failure and deterioration mechanism of the surrounding rock mass, the intermediate principal stresses $\sigma_2$ = 3MPa, 6MPa, 10MPa, and 14MPa are set. Table 1 and Fig 5 shows the loading path for details.

## Mechanical properties of coal and sandstone with different intermediate principal stress

### Characteristics of stress-strain curves of coal and sandstone

Fig 6 shows that the lateral strain of coal and sandstone under the action of different $\sigma_2$ is generally consistent. With the increase of $\sigma_2$, the growth rate of $\varepsilon_1$, $\varepsilon_2$, and $\varepsilon_3$ strains gradually decreases, and the lateral peak strain appears $\varepsilon_1$, $\varepsilon_2$ decreases, and $\varepsilon_3$ increases. The peak strain $\varepsilon_2$ of coal decreases by 54% from -18.9×10$^{-3}$ to -8.7×10$^{-3}$ from $\sigma_2$ = 3MPa to 14MPa, and the peak strain $\varepsilon_2$ of sandstone decreases by 32% from -16.1×10$^{-3}$ to -10.9×10$^{-3}$. The minimum principal strain $\varepsilon_3$ at the peak of coal increases by 93% from $\sigma_2$ = 3MPa to -15.1×10$^{-3}$, to -27.2×10$^{-3}$ at 14MPa, and sandstone increases by 59% from -14.3×10$^{-3}$ to -22.8×10$^{-3}$. The reason for the analysis is that the increase of the intermediate principal stress restricts the expansion deformation in the $\varepsilon_2$ direction, which leads to the gradual decrease of the lateral deformation in the $\varepsilon_2$ direction. The deformation of the rock is often generated along the weak force surface, and the increase of the intermediate principal stress forces the expansion phenomenon in the $\varepsilon_3$ direction.

In order to analyze the influence of $\sigma_2$ on the lateral peak strain of coal and sandstone, the lateral peak strain corresponding to different $\sigma_2$ was plotted (Fig 7). It can be seen from Fig 7 that with the increase of $\sigma_2$, the strain of coal and sandstone in the $\varepsilon_2$ direction decreases. The

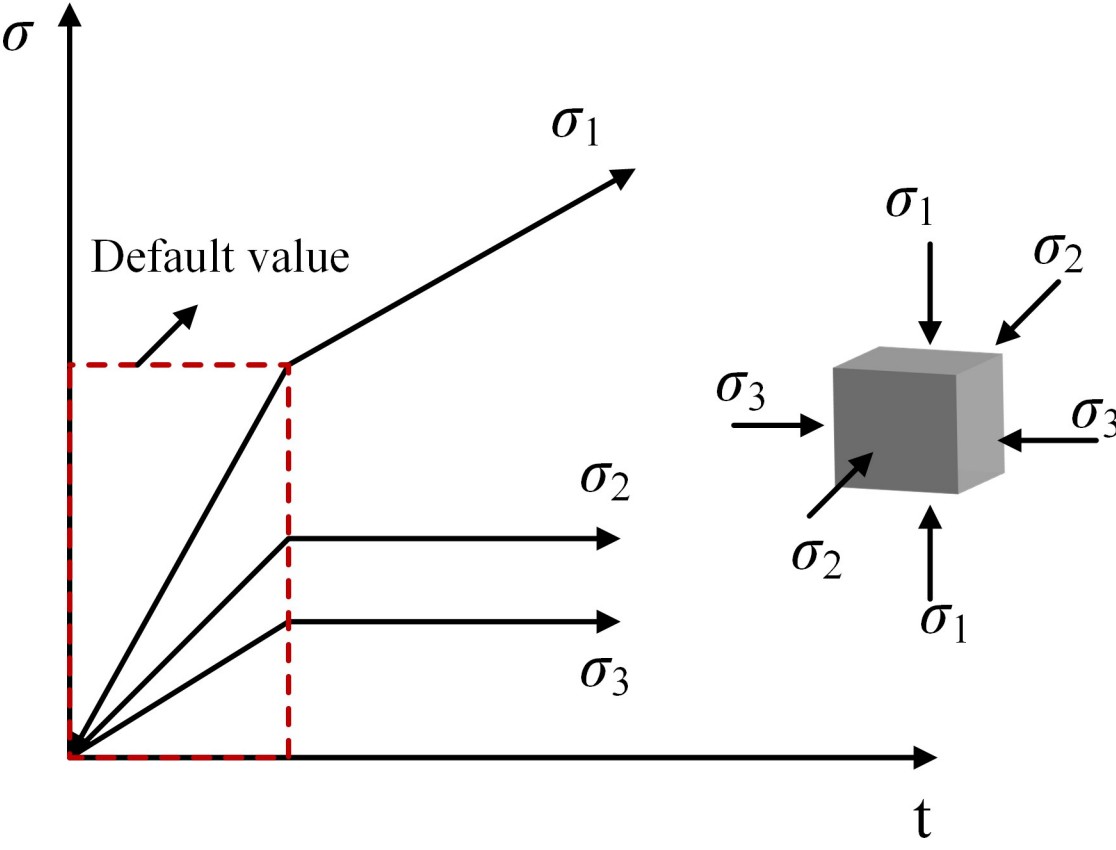

**Fig 5. Loading path.**

peak strain of coal and sandstone in the $\varepsilon_3$ direction increases, and the change of the peak strain of coal $\varepsilon_3$ is higher than that of sandstone due to the properties of the rock mass itself. The peak strain of coal at the peak of $\varepsilon_2$ is higher than that of sandstone before $\sigma_2 = 6$MPa, and the peak strain of coal is lower than that of sandstone after 6MPa. The reason for the analysis is that the coal body is not dense; there are many internal primary pores and fractures, and compared with the sandstone, it presents a complex structure, and the coal is prone to fractures and expansion under the action of the same lateral horizontal principal stress. With the development of fractures, the ability to connect the internal structures is further weakened, resulting in the aggravation of deformation. Compared with sandstone, the rate of fracture propagation under stress is slower, and the deformation is more limited, so the growth of lateral strain is less than that of coal.

## Mechanical characteristics of coal and sandstone

Because coal and rock roadways are two different lithology roadways, the bearing capacity of different rock masses is different under the same stress environment, leading to the instability and failure of a weak lithology body. Therefore, this paper discusses the difference of $\sigma_2$ in the strength failure characteristics of coal rock and sandstone through the octahedral strength criterion. Based on the Mises strength criterion, Mogi carried out many triaxial tests and proposed the octahedral strength criterion. The essence is still the shear failure criterion, which is essentially the relationship between the octahedral strength and the effective intermediate

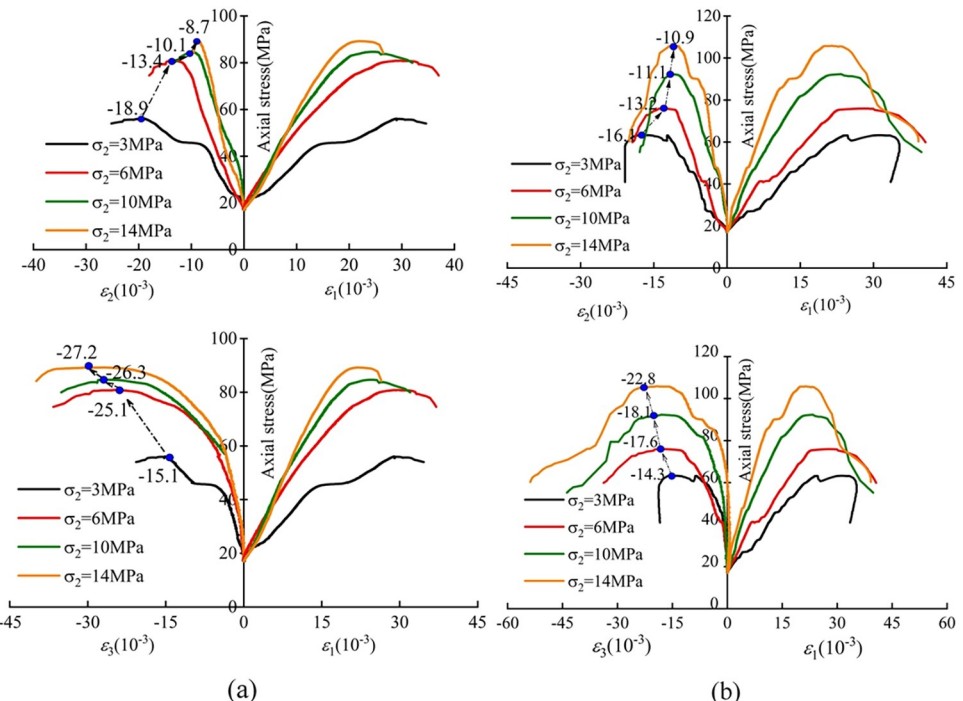

**Fig 6.** Lateral stress-strain curve of coal and sandstone Where (a) is coal, Where (b) is sandstone.

principal stress [24]. The formulas are as follows: (5), (6)

$$\tau_{oct} = \frac{1}{3} \sqrt{(\sigma_1 - \sigma_2)^2 + (\sigma_2 - \sigma_3)^2 + (\sigma_3 - \sigma_1)^2} \qquad (5)$$

$$\sigma_{m,2} = \frac{\sigma_1 + \sigma_3}{2} \qquad (6)$$

In the formula: $\tau_{oct}$ is octahedral shear stress, $\sigma_{m,2}$ is the effective intermediate principal stress, $\sigma_1$, $\sigma_2$, $\sigma_3$ are the maximum principal stress, intermediate principal stress, and minimum principal stress, respectively.

The relationship curve is drawn between the maximum shear stress and the effective intermediate principal stress. The strength characteristic curve of Fig 8 shows that the $\sigma_1$ of coal and sandstone increases with the increase of $\sigma_2$. From $\sigma_2 = 3$MPa to 14MPa, the peak strength of coal increases from 55.99MPa to 89.24MPa, which is enhanced by 37%. The peak strength of sandstone increases from 63.2MPa to 105.8MPa, which is enhanced by 40%. The influence of $\sigma_2$ on the peak strength of coal rock shows a power function relationship, while the sandstone has a linear relationship. The reason for the analysis is that the rock is not dense, there are many primary cracks and natural joints, and there are differences in the internal structure of the two kinds of rocks, which leads to a weaker effect of $\sigma_2$ on the strength of coal than that of sandstone. The strength of coal and sandstone obtained by the actual triaxial test is fitted by the above criteria. Considering the internal structure difference between coal and sandstone, the fitting coefficients reach 0.99421 (coal) and 0.99998 (sandstone), which are in good agreement with the existing true triaxial strength theory and test results of rock [25] and also verify the reliability and accuracy of the self-developed testing machine used in this paper.

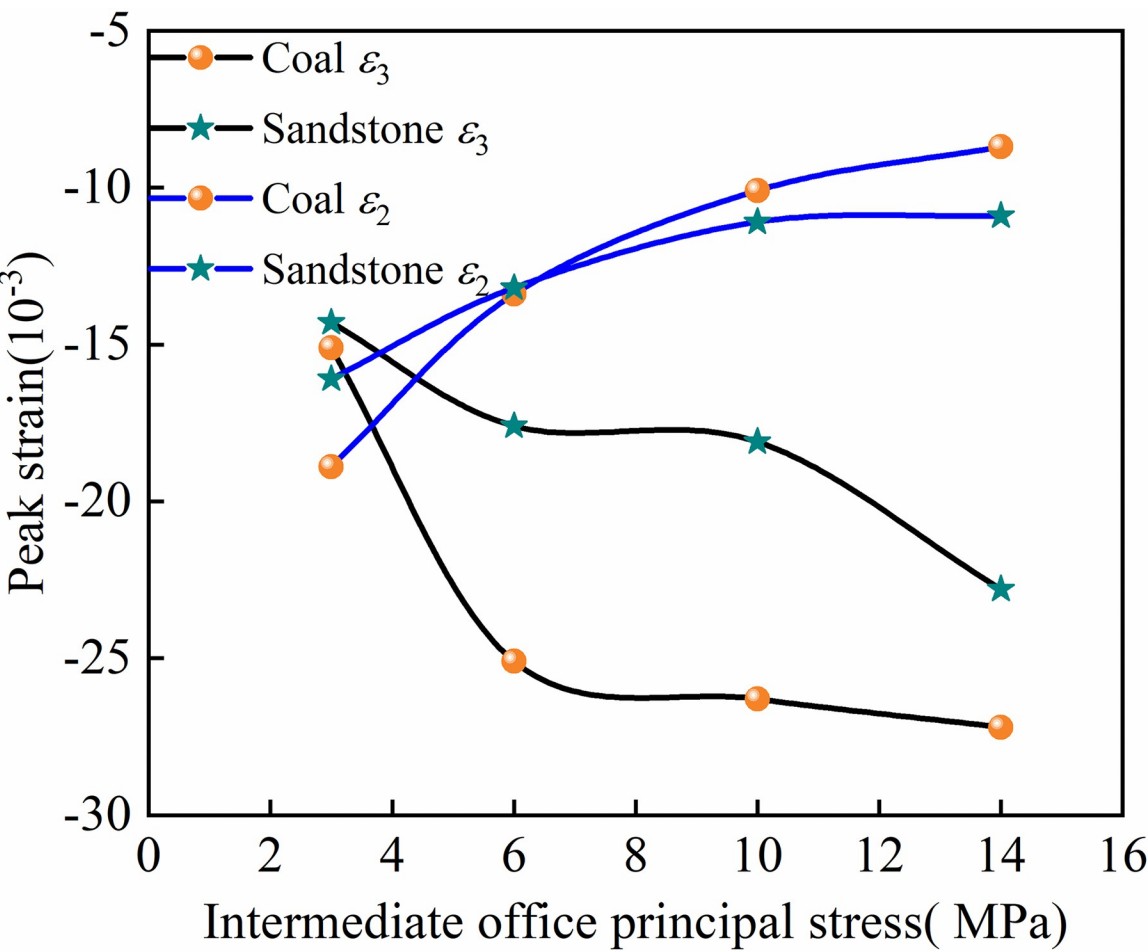

**Fig 7. Lateral peak strain of coal and sandstone.**

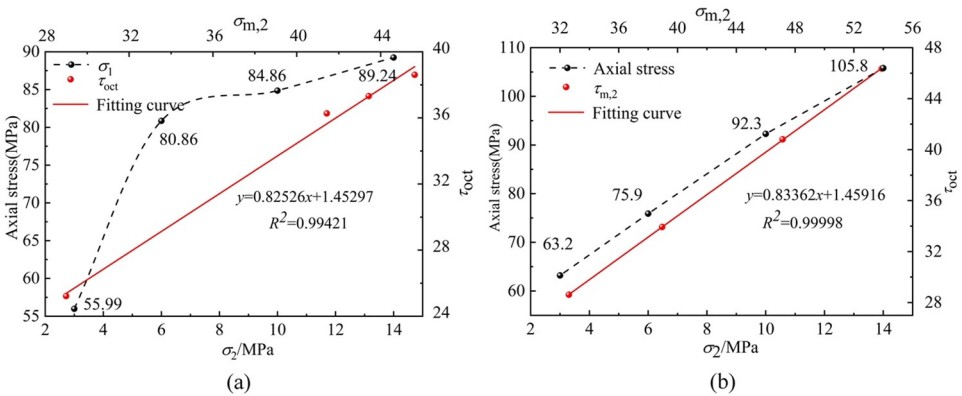

**Fig 8.** Fitting curve of strength characteristics of coal and sandstone Where (a) is coal, (b) is sandstone.

The elastic modulus is regarded as a measure of the difficulty of rock deformation. The larger the elastic modulus, the greater the ability to resist deformation. It can be seen from Fig 9 that the elastic modulus of coal and sandstone increases with the increase of $\sigma_2$. However, due to the influence of rock mass properties, the increase of the elastic modulus of sandstone is higher than that of coal. The minimum increase in the elastic modulus of sandstone is 0.3GPa, while the minimum increase in coal's elastic modulus is 0.09GPa. It can be seen that the sandstone structure is dense, the degree of bonding between the internal particles is high, and the change of $\sigma_2$ has little effect on the deformation of sandstone, showing a significant increase in its elastic modulus. However, coal is affected by the internal primary cracks and pore structure, and the degree of bonding between particles is much smaller than that of sandstone. Under the influence of $\sigma_2$, the occurrence state of primary crystals in coal is broken. With the increase of $\sigma_2$, the crystal chain moves to the weak surface, resulting in the gradual increase of strain and the decrease of elastic modulus of coal.

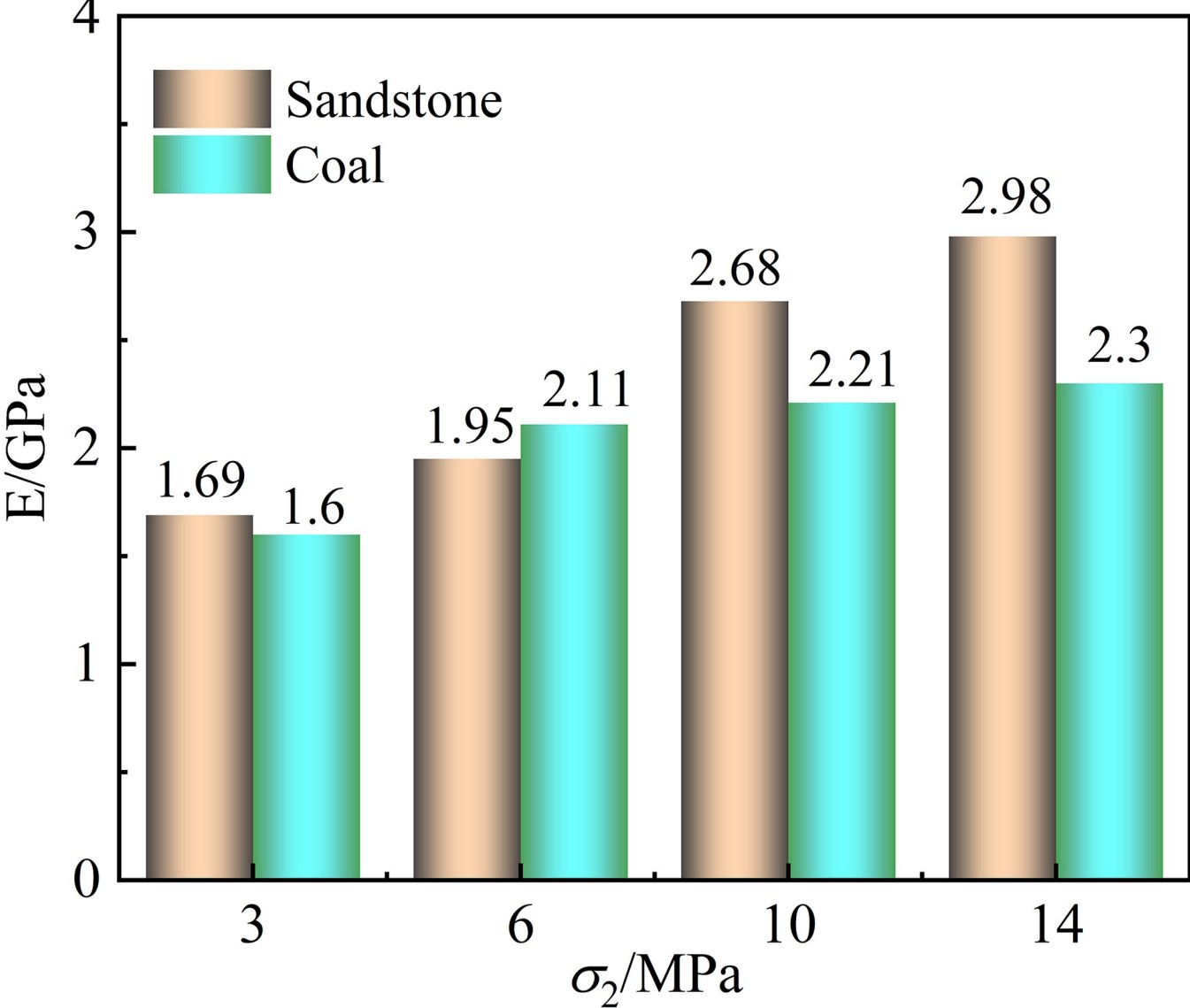

**Fig 9. Elastic modulus of coal and sandstone.**

## Macroscopic failure characteristics of coal and sandstone

Because the rock is a typical heterogeneous material, the failure mode is affected by the rock's physical properties and stress state. In the conventional triaxial test, the failure mode is the shear failure of low confining pressure and the plastic failure of high confining pressure. Under unequal stress in the accurate triaxial and triaxial directions, after the stress reaches the peak strength, a certain angle is formed between the direction parallel to $\sigma_2$ and the direction of $\sigma_3$. Different lithologies have different physical properties of internal micropores and components, and their destructive forms will differ.

From the failure characteristics of coal rock (a) and sandstone (b) in Fig 10, it can be seen that the failure mode of coal at the peak point is mainly a shear failure, while the failure mode of sandstone is a shear failure when $\sigma_2$ = 3MPa and 6MPa. After $\sigma_2$ = 10MPa, its failure mode changed and became tensile-shear composite failure dominated by tension. During the loading process, the lateral expansion of the rock is mainly along the $\sigma_3$direction, resulting in most of the internal crack development appearing on the $\sigma_1$-$\sigma_3$ plane and expanding along the $\sigma_2$direction. The more significant the difference between $\sigma_2$ and $\sigma_3$, the easier the tensile failure occurs in the $\sigma_3$ direction. The reason for the analysis is that the coal has low strength, heterogeneity, soft structure, and brittleness characteristics. With the increase of $\sigma_2$, the lateral deformation constraint effect of the coal is poor, the internal slip dislocation of the coal is more serious, the shear friction trace is more apparent, and the internal cracks are gradually developed. Finally, the macroscopic fracture surface produces more shear friction to overcome the limitation. After the sandstone is subjected to pressure in the other two directions, the displacement in the $\sigma_3$ direction will gradually increase, leading to the tensile strain in this direction exceeding the limit value, and the tensile failure characteristics will be shown when it is destroyed. It can be seen that the continuous increase of $\sigma_2$ will accelerates the formation and expansion of

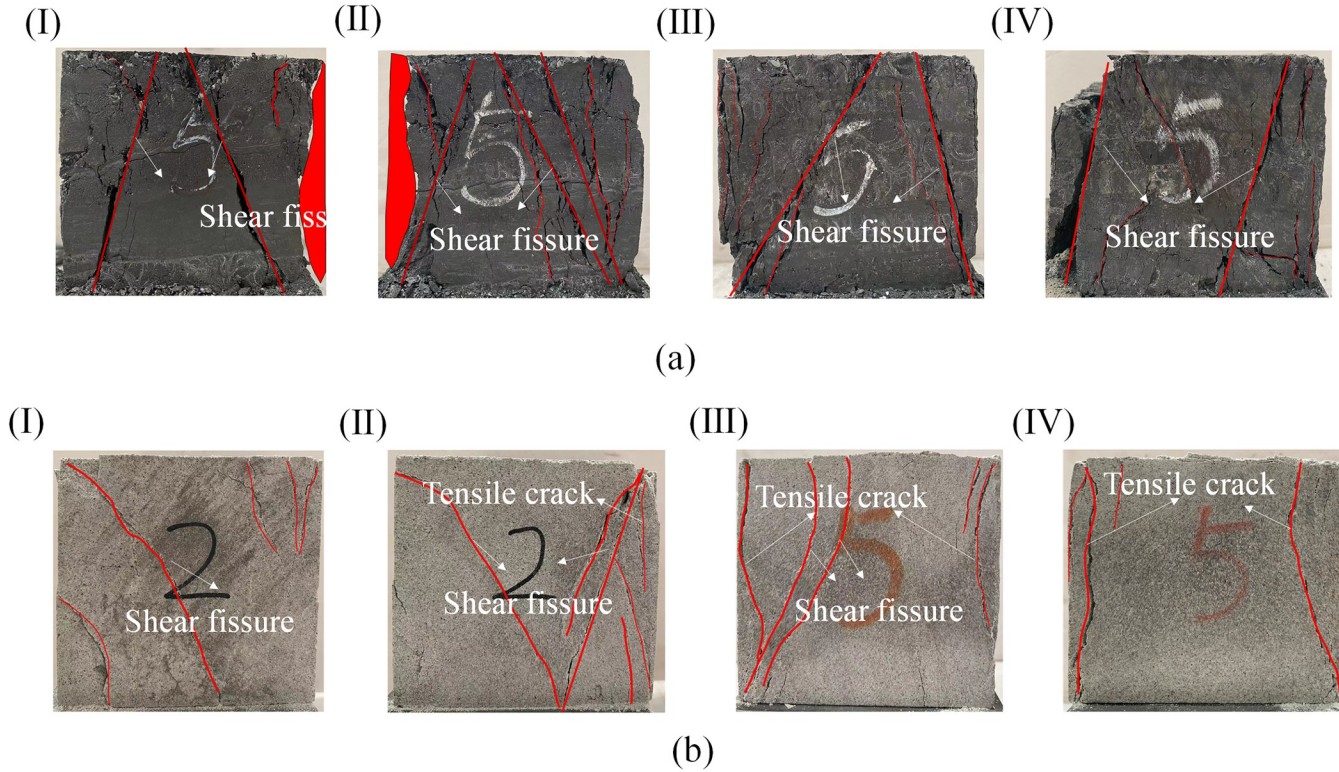

**Fig 10.** Peak failure diagram of coal and sandstone (a) is coal, (b) is sandstone (I, II, III, IV are $\sigma_2$ = 3MPa,6MPa,10MPa,14MPa).

tensile cracks in the rock, resulting in a decrease in rock strength and a significant decrease in the ability to withstand deformation.

## Acoustic emission characteristics of coal and sandstone with different intermediate principal stress

### Acoustic emission ringing count and cumulative energy

AE characteristics are derived from the changes in the interior of the material. Many micro-cracks, voids, and other micro-defects are often generated inside the rock. The acoustic emission information can reflect the development and evolution of cracks inside the rock under actual triaxial conditions and reveal the development and fracture mechanism of micro-cracks inside the rock [26, 27]. This paper describes the acoustic emission characteristics of coal and sandstone under different lithologies according to the influencing factors of different $\sigma_2$. The AE characteristics of coal and sandstone samples are analyzed from the perspective of axial stress, ringing count, and cumulative energy changing with time.

Fig 11 is the acoustic emission evolution law of coal and sandstone under different $\sigma_2$ under the continuous action of stress. There are micropores in the rock during the compaction stage. With the increase of stress, the internal pores gradually close so that the acoustic emission signal value is maintained at a low level. The cumulative energy growth rate of coal and sandstone is consistent, and a small amount of acoustic emission signal appears. In the elastic stage, the primary cracks in the rock have been closed, and some closed cracks have slipped, and the number and cumulative energy of acoustic emission show a slow growth. Due to the small constraint of the low intermediate principal stress on the lateral deformation, the rock is prone to micro-cracks, the acoustic emission signals of coal and sandstone are active, and the cumulative energy increase is significantly increased. In the plastic stage, due to the gradual development and penetration of cracks in the rock, the interaction between cracks intensifies, resulting in unstable growth of acoustic emission signals. When the rock is close to failure, the acoustic emission ringing count and cumulative energy increase abnormally. With the increase of $\sigma_2$, the degree of energy accumulation on the rock surface becomes increasingly apparent, and the acoustic emission signal intensity of coal and sandstone is enhanced when they are destroyed. However, sandstone's acoustic emission signal intensity is weaker than coal's.

By comparing the cumulative energy of acoustic emission in the same stress environment of different lithologies, it can be seen that during the stress loading process of coal, the cumulative energy curve has no ladder shape, showing a gradual growth trend, while during the stress loading process of sandstone, the cumulative energy curve has apparent ladder shape. The reason for the analysis is that the compactness of coal is much weaker than sandstone's. When cracks appear inside, the cumulative energy increases significantly. The interaction between the internal particles of sandstone with better compactness plays a role in the undertaking, making sandstone maintain its overall stability. The internal weak structure, large pores, and other factors do not have this ability for coal, so the cumulative energy curve gradually increases. Before the coal and sandstone are on the verge of failure, the abnormal sudden increase of the acoustic emission ringing count, the high frequency of cumulative energy, and the sharp rise indicate that the fracture inside the rock is fully pregnant and expanded, which is a crucial precursor signal for the imminent instability and failure of the rock.

### RA-AF value characteristics of coal and sandstone

The acoustic emission parameters of micro-fracture in rock can reflect the fracture properties of the acoustic emission source. To further study the fracture characteristics of rock and

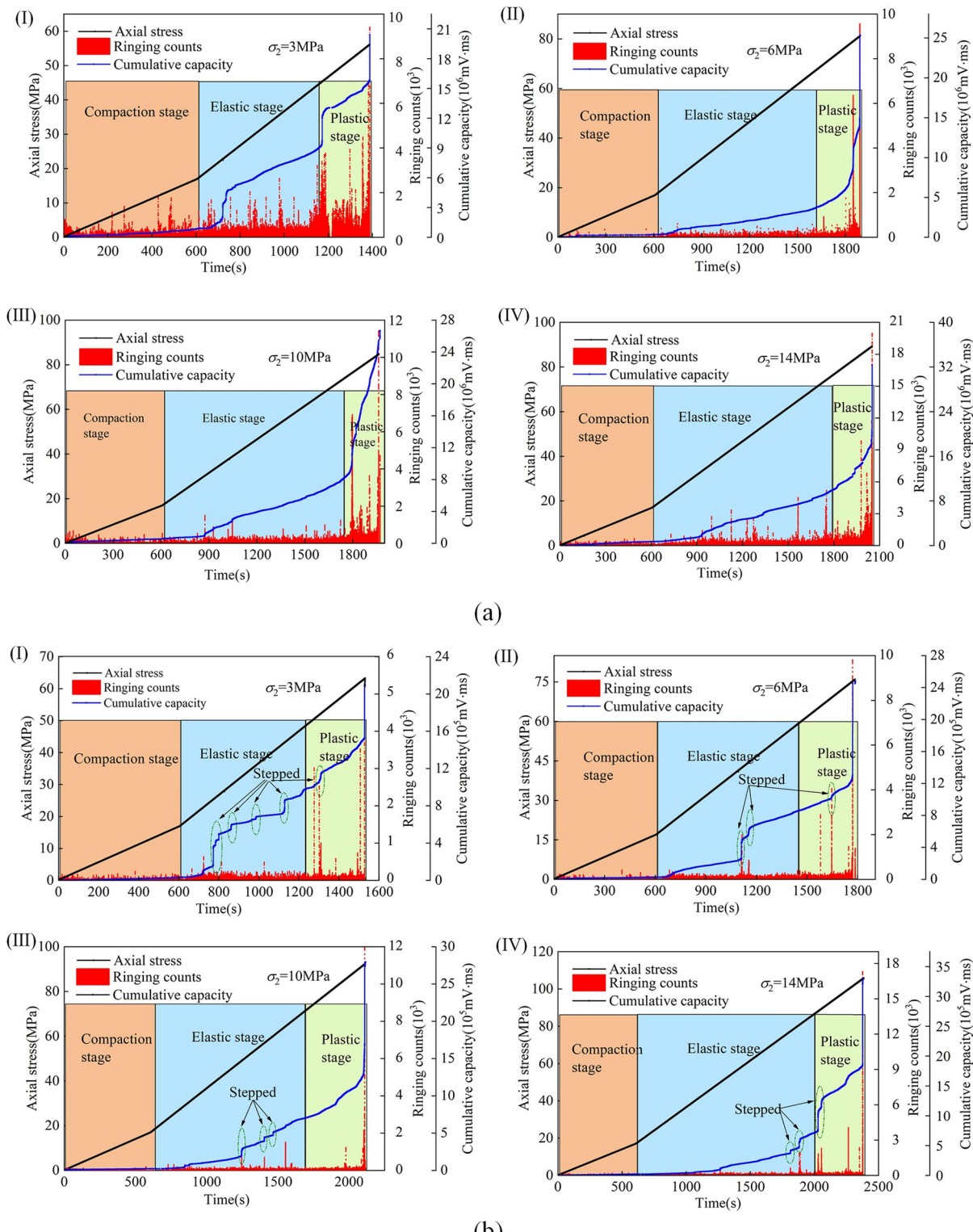

**Fig 11.** AE evolution law of coal and sandstone (a) is coal, (b) is sandstone (I, II, III, IV are $\sigma_2$ = 3MPa,6MPa,10MPa,14MPa).

analyze the fracture mechanism of coal and sandstone under different $\sigma_2$ conditions, the *RA* value and average frequency *AF* value in acoustic emission parameters describe the micro-fracture type of rock. The study shows that the *RA* value and *AF* value in the acoustic emission parameters can describe the fracture type of the rock sample [28]. There are two main modes of micro-cracks in the failure process of rock samples: tension and shear. The *RA* value and *AF* value in acoustic emission are used to judge the crack failure characteristics of rock during loading. The calculation method is shown in the following formulas (7) and (8) [29, 30]:

$$AF = \frac{AE\ ringing\ counts}{Duration\ time} \tag{7}$$

$$RA = \frac{Rise\ time}{Maximum\ amplitude} \tag{8}$$

When the two parameters of *RA* value and *AF* value are summarized according to many waveform signals, for tensile cracks, due to the instantaneous release of elastic energy, the rise time and duration are short, the amplitude is large, and the ringing count is large. At this time, *RA* is low, *AF* is high, and shear cracks are opposite. *RA* value and *AF* value are particular means to analyze fracture development [31]. Many scholars have deeply explored the characteristics of *RA-AF* and proposed *RA-AF* characteristics to describe the characteristics of rock mass crack development. The slope of the dividing line in Fig 12 is defined as *k*. When *AF/RA>k*, the crack shows tensile failure; when *AF/RA <k*, the crack shows shear failure.

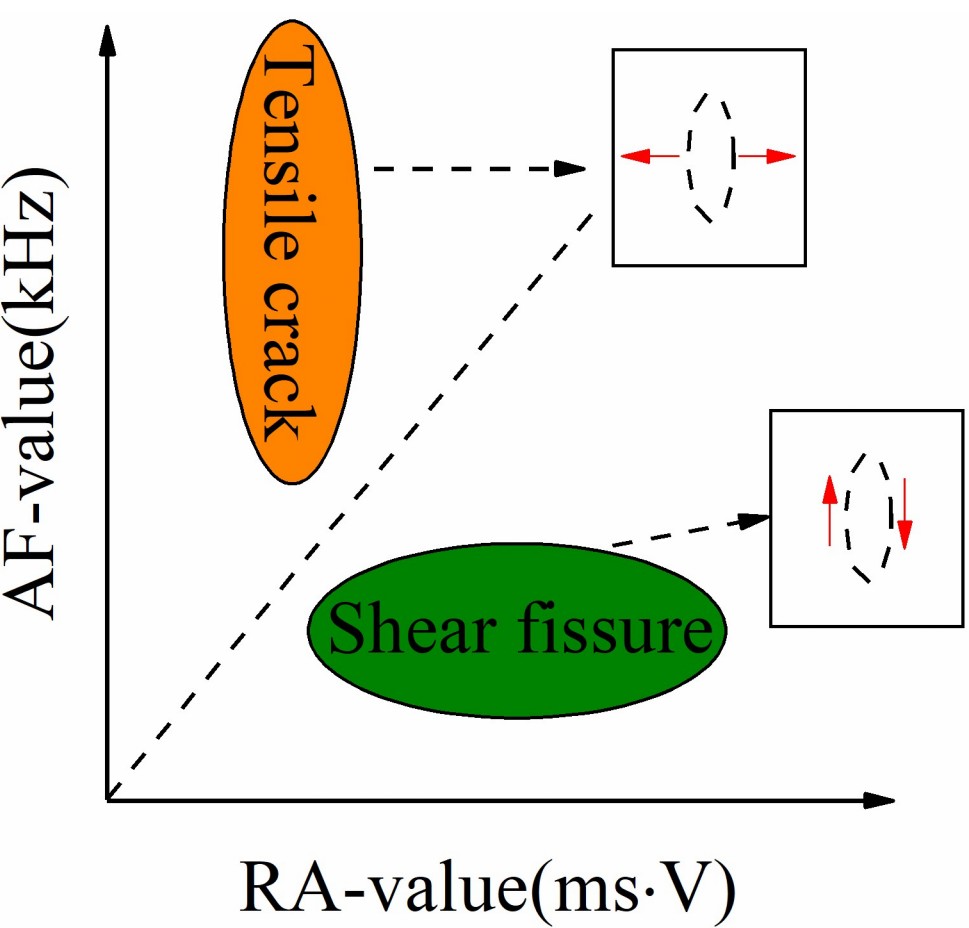

**Fig 12. RA-AF value relationship diagram.**

**Table 2. Statistics of the proportion of tensile and shear cracks in coal and sandstone.**

| $\sigma_2$/MPa | Elastic stage crack | | | | Plastic stage crack | | | | Total crack | | | |
|---|---|---|---|---|---|---|---|---|---|---|---|---|
| | Shear ratio% | | Percentage share% | | Shear ratio% | | Percentage share% | | Shear ratio% | | Percentage share% | |
| | Coal | sandstone | Coal | sandstone | Coal | sandstone | Coal | sandstone | Coal | sandstone | Coal | sandstone |
| 3 | 62.57 | 77.26 | 29.79 | 22.74 | 66.42 | 75.54 | 33.58 | 24.46 | 67.99 | 75.62 | 32.01 | 24.38 |
| 6 | 71.94 | 58.72 | 28.06 | 41.28 | 69.74 | 54.99 | 30.26 | 45.01 | 70.73 | 55.80 | 29.27 | 44.20 |
| 10 | 68.95 | 35.45 | 31.05 | 64.55 | 68.18 | 29.24 | 31.85 | 70.76 | 68.47 | 20.81 | 31.53 | 69.19 |
| 14 | 74.67 | 33.81 | 25.33 | 66.19 | 74.69 | 43.15 | 25.31 | 56.85 | 71.10 | 38.24 | 28.90 | 61.76 |

At present, most scholars do not have the same value when taking k; Wu et al. [32] used $k = 1/60$ for comparison; Ohtsu [33] was compared using $k = 0.1$; In uniaxial compression, Liu [34] took $k$ as 14.3, 75, 53.8 and 15.8 for granite, limestone, mudstone, and sandstone, respectively, and $k = 0.1$ for *RA-AF* comparison. In order to further study the characteristics of rock fracture, the proportion of cracks in the elastic stage and plastic stage was counted by the proportion of *RA-AF* value and the proportion of cracks in the whole process. It can be concluded that shear cracks dominate the internal crack propagation of coal. In contrast, the development of internal cracks in sandstone is dominated by shear failure with the increase of $\sigma_2$, and the specific parameters are shown in Table 2.

The *RA-AF* crack development diagram of coal is shown in Fig 13. In the elastic stage, when the intermediate principal stress $\sigma_2 = 3$MPa, the distribution of internal cracks in coal shows that the proportion of shear cracks is 70.21%, and the proportion of tensile cracks is 29.72%. When the intermediate principal stress $\sigma_2 = 14$MPa, the proportion of shear cracks is 74.67%, and the proportion of tensile cracks is 25.33%. In the plastic stage, when $\sigma_2 = 3$MPa, the distribution of internal cracks in coal shows that shear cracks account for 66.42% and tensile cracks account for 33.58%. When $\sigma_2 = 14$MPa, shear cracks account for 74.69%, and tensile cracks account for 25.31%. The proportion of shear cracks gradually increases with the increase of $\sigma_2$. The reason for the analysis is that the increase of $\sigma_2$ has a lateral constraint effect on coal and rock, which makes the fracture gradually overcome the confining pressure shear constraint, and the increase of $\sigma_2$ gradually inhibits the slip of the fracture.

The *RA-AF* crack development diagram of sandstone is shown in Fig 14. When the intermediate principal stress is low; the internal shear crack dominates in the elastic stage. With the increase of $\sigma_2$, the shear crack gradually decreases, and the tensile crack begins to appear. When $\sigma_2 = 14$MPa, the proportion of internal shear crack is 33.81%, and the proportion of tensile crack is 66.19%. In the process of the plastic stage, the internal shear crack still dominates when $\sigma_2 = 14$MPa and the proportion of internal cracks begins to change with the increase of intermediate principal stress $\sigma_2$, from a low-stress shear crack to a high-stress tensile crack. The reason for the analysis is that when the intermediate principal stress $\sigma_2$ is high, the sandstone is dominated by tensile crack propagation, resulting in the rapid intersection of tensile cracks in the sandstone, resulting in the instantaneous formation of the macroscopic fracture surface (tension), which also explains the phenomenon of ' sudden increase ' in the number of cumulative acoustic emission events.

## Discussion

Based on the stress-strain law and acoustic emission characteristics of coal and sandstone under accurate triaxial loading, the similarities between coal and sandstone under the same stress environment are found: the increase of $\sigma_2$ inhibits the lateral deformation of rock and improves the peak strength and elastic modulus of coal and sandstone. When the intermediate

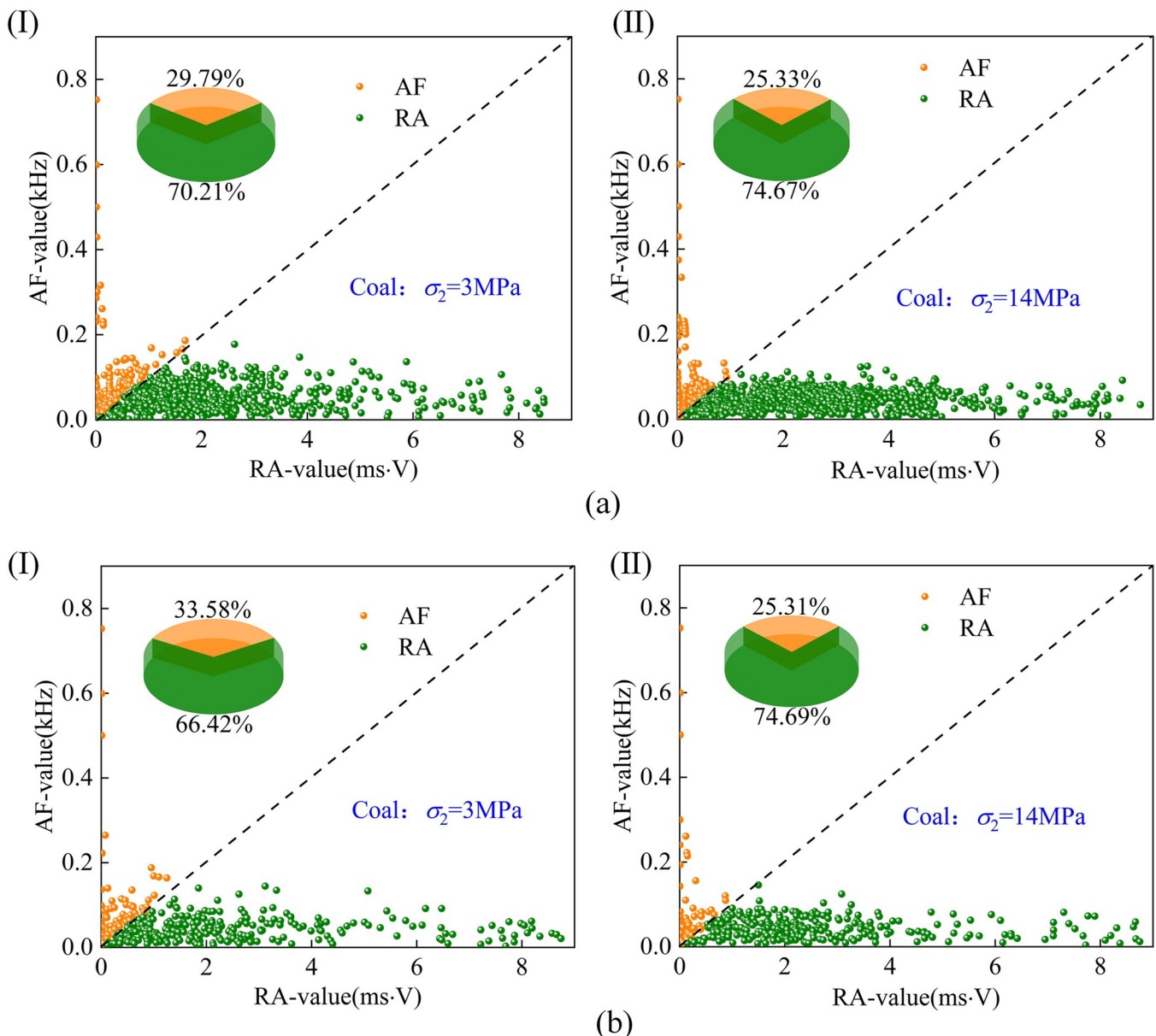

**Fig 13.** Coal RA-AF crack proportion (a) elastic stage, (b) plastic stage(I,II are $\sigma_2$ = 3MPa,14MPa).

principal stress is low, the acoustic emission signal in the elastic stage increases abnormally; when the peak is damaged, the acoustic emission signal (ring count and cumulative energy) increases with the increase of $\sigma_2$.

The difference is that the increase of sandstone's peak strength and elastic modulus is higher than that of coal. The failure mode of coal shows shear failure, and the sandstone changes from shear failure with low intermediate principal stress to tensile-shear composite failure with high intermediate principal stress. The cumulative energy in the acoustic emission of sandstone appears to be a step-like phenomenon.

The reason for the different phenomena is that coal is transformed from plant remains by biochemical action, which is brittle, has many primary cracks, and complex structures. Various

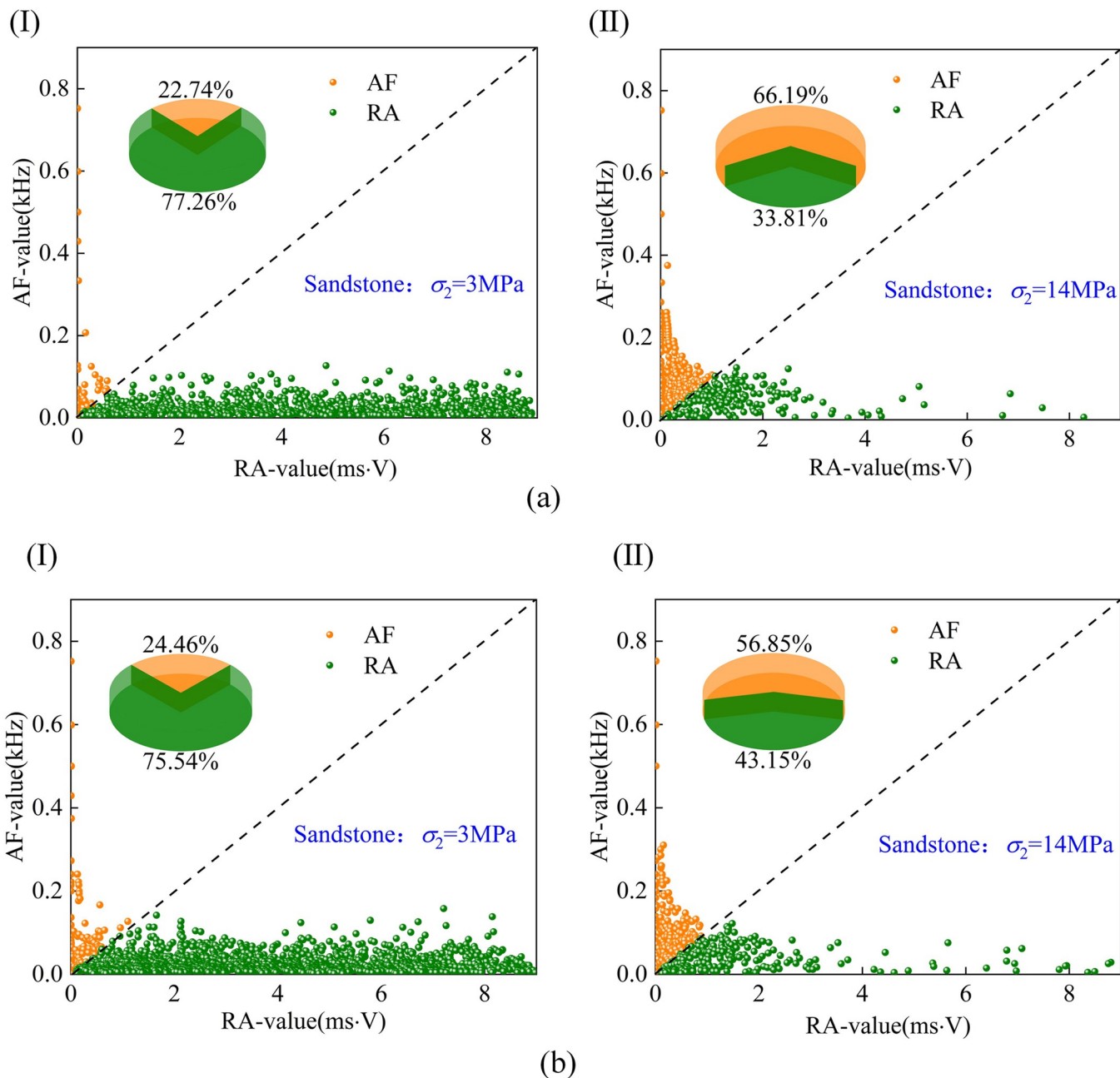

**Fig 14.** Sandstone RA-AF crack proportion (a) elastic stage, (b) plastic stage(I,II are $\sigma_2$ = 3MPa,14MPa).

sand particles cement sandstone, and its internal structure is stable. Therefore, there will be some differences between the two lithologic rocks under the same stress environment. It is worth noting that a single lithologic roadway's overall stability and final strength depend on itself. In contrast, a semi-coal rock roadway's stability and peak strength depend on weak lithologic rock mass. Due to the weak bonding force between coal and sandstone, shear dislocation is formed at the boundary of weak structural planes under the action of roof and excavation, resulting in shear failure(Fig 15). Moreover, the deformation of the weak structure (coal structure) at the two sides of the roadway is the largest, which makes the two sides of the roadway

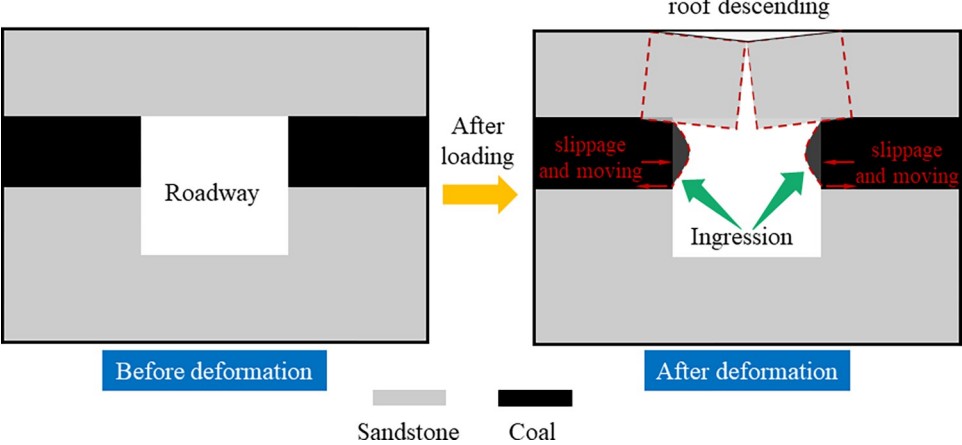

**Fig 15. Semi-coal rock roadway schematic diagram.**

move inward and the deformation phenomenon more serious. With the increase of buried depth, the deformation phenomenon of the roadway will be more obvious. To ensure the stability of the surrounding rock of the semi-coal rock roadway, the partition grouting reinforcement control method can be adopted, and the high pre-stressed bolt (cable)+ anchor net + grouting reinforcement arrangement for the weak rock mass is the primary means to achieve the stability of the surrounding rock of the roadway.

## Conclusion

In this paper, the theory of deviator stress is introduced to analyze the influence of deviator stress on the plastic zone of the roadway. A triaxial testing machine studies the mechanical properties and acoustic emission evolution law of coal and sandstone with different intermediate principal stresses. The standard and different characteristics of different lithologies under the same stress are analyzed, and the following conclusions are obtained:

1. The increase of $\sigma_2$ inhibits the development of strain in the $\sigma_2$ direction of coal and sandstone, promotes the increase of strain in the $\sigma_3$ direction, and gradually increases the induced lateral strain difference. Its peak strength and elastic modulus are positively correlated with $\sigma_2$. However, the amplification effect is different; the peak strength and elastic modulus of sandstone increase by 40% and 76% from $\sigma_2 = 3MPa$ to $\sigma_2 = 14MPa$, respectively, while the peak strength and elastic modulus of coal increase by 37% and 44%, respectively.

2. The ringing count and cumulative energy generated by the acoustic emission at the peak of coal and sandstone are positively correlated with $\sigma_2$. Due to the different lithologies, the coal acoustic emission ringing count occurs frequently, the value is significant, and the cumulative energy of the acoustic emission is low. In contrast, the sandstone acoustic emission ringing count is small; the value is small. The cumulative energy of the acoustic emission is significant, and the cumulative energy of the coal acoustic emission gradually increases without the ladder phenomenon. The cumulative energy of sandstone acoustic emission appears to have many apparent steps.

3. With the increase of $\sigma_2$, there are apparent differences in the failure characteristics of different lithologies under the same $\sigma_2$. The proportion of *RA* value of sandstone decreased by 37.38% from 75.62% to 38.24%, and the failure characteristics changed from the primary

shear failure to the tensile and shear composite failure mode. However, with the increase of $\sigma_2$, the development of shear cracks was intensified, and the proportion of *RA* value increased by 3.11% from 67.99% to 71.10%. The characteristics of shear failure became more and more evident during failure.

4. Under the same stress environment, there are apparent differences in the mechanical properties and crack development of different lithology rocks. When the roadway is a semi-coal rock roadway, the roadway often fails in the weak rock mass. At this time, the weak rock mass is strengthened. Support and weaken the risk of uncoordinated deformation and failure due to different lithological rock masses to enhance the overall stability of the surrounding rock of the roadway.

## Author Contributions

**Conceptualization:** Wenbao Shi, Qingzhao Xu, Chao Qi, Chuanming Li.

**Investigation:** Qingzhao Xu, Zhuang Miao, Aoyun Yan.

**Project administration:** Wenbao Shi, Jucai Chang.

**Writing – original draft:** Qingzhao Xu.

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
