## [Decision Letter · Decision Letter 0]

30 Sep 2024

PONE-D-24-40527Experimental study on mechanical response and crack evolution law of coal and sandstone under different stress environmentsPLOS ONE

Dear Dr. Xu,

Thank you for submitting your manuscript to PLOS ONE. After careful consideration, we feel that it has merit but does not fully meet PLOS ONE’s publication criteria as it currently stands. Therefore, we invite you to submit a revised version of the manuscript that addresses the points raised during the review process.

We look forward to receiving your revised manuscript.

Kind regards,

Xianggang Cheng, Ph.D.

Academic Editor

PLOS ONE

Journal Requirements: When submitting your revision, we need you to address these additional requirements. 1. Please ensure that your manuscript meets PLOS ONE's style requirements, including those for file naming. The PLOS ONE style templates can be found at https://journals.plos.org/plosone/s/file?id=wjVg/PLOSOne_formatting_sample_main_body.pdf and https://journals.plos.org/plosone/s/file?id=ba62/PLOSOne_formatting_sample_title_authors_affiliations.pdf 2. Thank you for stating the following financial disclosure: "This work was supported by the National Natural Science Foundation of China (Grant Nos.52104117-SWB and 52174105-CJC and 52174103-LCM) and Excellent Scientific Research and Innovation Team (2023AH010023)" Please state what role the funders took in the study.  If the funders had no role, please state: ""The funders had no role in study design, data collection and analysis, decision to publish, or preparation of the manuscript."" If this statement is not correct you must amend it as needed. Please include this amended Role of Funder statement in your cover letter; we will change the online submission form on your behalf. 3. When completing the data availability statement of the submission form, you indicated that you will make your data available on acceptance. We strongly recommend all authors decide on a data sharing plan before acceptance, as the process can be lengthy and hold up publication timelines. Please note that, though access restrictions are acceptable now, your entire data will need to be made freely accessible if your manuscript is accepted for publication. This policy applies to all data except where public deposition would breach compliance with the protocol approved by your research ethics board. If you are unable to adhere to our open data policy, please kindly revise your statement to explain your reasoning and we will seek the editor's input on an exemption. Please be assured that, once you have provided your new statement, the assessment of your exemption will not hold up the peer review process. 4. In the online submission form, you indicated that "The datasets used and/or analyzed during the current study available from the corresponding author on reasonable request" All PLOS journals now require all data underlying the findings described in their manuscript to be freely available to other researchers, either 1. In a public repository, 2. Within the manuscript itself, or 3. Uploaded as supplementary information.This policy applies to all data except where public deposition would breach compliance with the protocol approved by your research ethics board. If your data cannot be made publicly available for ethical or legal reasons (e.g., public availability would compromise patient privacy), please explain your reasons on resubmission and your exemption request will be escalated for approval. 5. Please upload a copy of Figure 1-14, to which you refer in your text on page 5-18. If the figure is no longer to be included as part of the submission please remove all reference to it within the text.

**Additional Editor Comments:**

Corresponding stress condition evidence should be supplemented to further support the manuscript; More recent references should be added; When resubmitting, please insert the image into the manuscript.

Reviewers' comments:

Reviewer's Responses to Questions

**Comments to the Author**

1. Is the manuscript technically sound, and do the data support the conclusions?

Reviewer #1: Yes

Reviewer #2: Yes

2. Has the statistical analysis been performed appropriately and rigorously? 

Reviewer #1: Yes

Reviewer #2: Yes

3. Have the authors made all data underlying the findings in their manuscript fully available?

Reviewer #1: Yes

Reviewer #2: Yes

4. Is the manuscript presented in an intelligible fashion and written in standard English?

Reviewer #1: Yes

Reviewer #2: Yes

5. Review Comments to the Author

Reviewer #1: This paper studies the stability of the roadway under different stress environments and lithological conditions. The mechanical characteristics of coal rock with different intermediate principal stresses are studied by theoretical analysis and actual triaxial experiments. The influence of the change of intermediate principal stress on different lithologic materials is analyzed. The influence characteristics of the change of intermediate principal stress are obtained, and the support mode of the semi-coal rock roadway is proposed to improve the stability of the roadway. After review and study, it has a certain degree of innovation, but the article still has the following problems:

Problems 1: The summary section suggests further condensation.

Problems 2: What is the basis for the design of the stress field in the test? How are the numbers considered? Please explain.

Problems 3: The analysis of Figure 6 is not thorough enough, and the main reasons for the different changes between the two need to be further analyzed.

Problems 4: It is proposed that sections 4.1 and 4.3 be combined to reorganize the analysis.

Problems 5: It is mentioned in many places in the article that with the increase of the second principal stress, it is too verbose, and it is recommended to delete the irrelevant text and re-summarize the conclusion.

Problems 6: Starting from the stress field of the surrounding rock, the stress state of the rock mass under the actual triaxial stress state is introduced, and the clarification sequence is reasonable and smooth, but the literature in the past five years accounts for a small proportion of the references, so it is recommended to update further.

Reviewer #2: Authors have conducted experimental study on mechanical response and crack evolution law of coal and sandstone under different stress environments. Before the manuscript can be recommended to be accepted, authors have to make some major revisions as follows:

1) Language and grammar errors:

L39 “presents the complex stress characteristics”, this sentence lacks a subject.

L77 “…for acoustic emission Chen…”

Please check the language grammar used in the whole manuscript carefully to avoid these errors. It would be helpful to do the language polishing by a native language speaker.

2) In section Experimental scheme

a) More information about research objects should be provided such as sampling location of rock, basic properties and so on.

b) Why the default value of the maximum principal stress σ1 is 17MPa and the minimum principal stress σ3 is 2MPa? How are the four different values of intermediate principal stress selected? It is suggested supply experiment design basis in this section.

3) In L211, it would be good to supply references to prove this research results are in good agreement with the existing research.

4) Conclusions: it is suggested to add one more paragraph introducing the methodology briefly employed in this work, before drawing specific conclusions.

6. PLOS authors have the option to publish the peer review history of their article (what does this mean?). If published, this will include your full peer review and any attached files.

Reviewer #1: No

Reviewer #2: No

---

## [Author Response · Author response to Decision Letter 0]

10 Oct 2024

Replies to reviewer #1:

This paper studies the stability of the roadway under different stress environments and lithological conditions. The mechanical characteristics of coal rock with different intermediate principal stresses are studied by theoretical analysis and actual triaxial experiments. The influence of the change of intermediate principal stress on different lithologic materials is analyzed. The influence characteristics of the change of intermediate principal stress are obtained, and the support mode of the semi-coal rock roadway is proposed to improve the stability of the roadway. After review and study, it has a certain degree of innovation, but the article still has the following problems:

(1) The summary section suggests further condensation.

Thank you for your valuable advice. In response to your suggestion to refine the abstract, we have summarized the description and marked it with a yellow background in the revised version. The specific changes are as follows:

In order to study the mechanical response and crack evolution law of different lithologic rock bodies under different stress environments in deep stress mines, based on the deviator stress theory, the actual triaxial disturbance unloading rock test system was used to simulate the stress occurrence environment of the original rock. The mechanical characteristics of different σ2 coal rock masses were studied, and the crack evolution law of coal and sandstone under different stress environments was analyzed. The results show that the increase of σ2 inhibits the deformation in the σ2 direction of coal and sandstone, promotes the expansion and deformation in the σ3 direction, and enhances its peak strength and elastic modulus. The development characteristics of internal cracks in rock mass are directly related to the stress environment, and the increase of σ2 promotes the increase of the proportion of coal RA value, weakens the proportion of sandstone RA value, aggravates the development of coal internal shear cracks, and inhibits the development of internal shear cracks in sandstone. The larger σ2, the greater the initial AE ringing count of coal and sandstone, and the greater the AE cumulative energy when the rock mass is finally damaged. At the same time, due to the self-organizing behavior in the process of crystal failure in sandstone, the cumulative energy curve of sandstone fluctuates in a step-like manner. The ringing count and cumulative energy increase suddenly, which can predict the imminent instability and failure of the rock, and the research results can provide an experimental basis for the sudden instability of deep high-stress roadways.

(2) What is the basis for the design of the stress field in the test? How are the numbers considered? Please explain.

Thank you for your advice. In response to your question about how the stress field is set in the article, we have made additional changes in the revised manuscript and marked it with a yellow background. For specific changes, please refer to the section of the experimental protocol. The specific changes are as follows:

According to the schematic diagram of the stress distribution of the surrounding rock of the roadway (Fig.4), it can be seen that due to the change of the principal stress in the process of excavation of the roadway, as the rock mass is far away from the free side, the horizontal stress shows a transition from the direction of stress increase - stress reduction - original stress, at this time, the maximum principal stress is the vertical principal stress (considering the buried depth is 680m, the vertical principal stress σ1=17MPa), the position of the unit body is selected in the roadway gang and near the free side, so the minimum principal stress σ3=2MPa is set, and at the same time, The rock mass at the roadway excavation face has no supporting effect, which forces the rock mass in front of it to produce slow deformation behavior, resulting in changes in the horizontal stress at different positions of the rock mass in front of the roadway (intermediate principal stress σ2). Therefore, to study the influence of different intermediate principal stresses on the failure and deterioration mechanism of the surrounding rock mass, the intermediate principal stresses σ2=3MPa, 6MPa, 10MPa, and 14MPa are set.

Fig.4 Schematic diagram of stress distribution in surrounding rock of roadway

(3) The analysis of Figure 6 is not thorough enough, and the main reasons for the different changes between the two need to be further analyzed.

Thank you for your advice. In response to your question, we have performed a redescription analysis and marked the revised manuscript with a yellow background. The specific changes are as follows:

In order to analyze the influence of σ2 on the lateral peak strain of coal and sandstone, the lateral peak strain corresponding to different σ2 was plotted (Fig.7). It can be seen from Fig.7 that with the increase of σ2, the strain of coal and sandstone in the ε2 direction decreases. The peak strain of coal and sandstone in the ε3 direction increases, and the change of the peak strain of coal ε3 is higher than that of sandstone due to the properties of the rock mass itself. The peak strain of coal at the peak of ε2 is higher than that of sandstone before σ2 =6MPa, and the peak strain of coal is lower than that of sandstone after 6MPa. The reason for the analysis is that the coal body is not dense; there are many internal primary pores and fractures, and compared with the sandstone, it presents a complex structure, and the coal is prone to fractures and expansion under the action of the same lateral horizontal principal stress. With the development of fractures, the ability to connect the internal structures is further weakened, resulting in the aggravation of deformation. Compared with sandstone, the rate of fracture propagation under stress is slower, and the deformation is more limited, so the growth of lateral strain is less than that of coal.

(4) It is proposed that sections 4.1 and 4.3 be combined to reorganize the analysis. 

Thank you for your advice. In response to your question, we make the following explanations: in the analysis of the experimental results, we first analyze the stress-strain curves of coal and sandstone to obtain the characteristics of the stress-strain curves of the two, secondly, according to their peak strength and elastic modulus, obtain the response relationship between the peak strength and elastic modulus of the two and the intermediate principal stress, and finally comprehensively sort out the fundamental mechanical characteristics of the stress-strain curve, peak strength and elastic modulus—a more comprehensive analysis of the main factors that led to the macro disruption of both. Therefore, the logical order of the articles will be more precise by analyzing them separately.

(5) It is mentioned in many places in the article that with the increase of the second principal stress, it is too verbose, and it is recommended to delete the irrelevant text and re-summarize the conclusion.

Thank you for your advice. In response to your question, we have made changes in the revised manuscript and marked the changes with a yellow background. The specific changes are as follows:

In this paper, the theory of deviator stress is introduced to analyze the influence of deviator stress on the plastic zone of the roadway. A triaxial testing machine studies the mechanical properties and acoustic emission evolution law of coal and sandstone with different intermediate principal stresses. The standard and different characteristics of different lithologies under the same stress are analyzed, and the following conclusions are obtained:

(1) The increase of σ2 inhibits the development of strain in the σ2 direction of coal and sandstone, promotes the increase of strain in the σ3 direction, and gradually increases the induced lateral strain difference. Its peak strength and elastic modulus are positively correlated with σ2. However, the amplification effect is different; the peak strength and elastic modulus of sandstone increase by 40% and 76% from σ2=3MPa to σ2=14MPa, respectively, while the peak strength and elastic modulus of coal increase by 37% and 44%, respectively.

(2) The ringing count and cumulative energy generated by the acoustic emission at the peak of coal and sandstone are positively correlated with σ2. Due to the different lithologies, the coal acoustic emission ringing count occurs frequently, the value is significant, and the cumulative energy of the acoustic emission is low. In contrast, the sandstone acoustic emission ringing count is small; the value is small. The cumulative energy of the acoustic emission is significant, and the cumulative energy of the coal acoustic emission gradually increases without the ladder phenomenon. The cumulative energy of sandstone acoustic emission appears to have many apparent steps.

(3) With the increase of σ2, there are apparent differences in the failure characteristics of different lithologies under the same σ2. The proportion of RA value of sandstone decreased by 37.38% from 75.62% to 38.24%, and the failure characteristics changed from the primary shear failure to the tensile and shear composite failure mode. However, with the increase of σ2, the development of shear cracks was intensified, and the proportion of RA value increased by 3.11% from 67.99% to 71.10%. The characteristics of shear failure became more and more evident during failure.

(4) Under the same stress environment, there are apparent differences in the mechanical properties and crack development of different lithology rocks. When the roadway is a semi-coal rock roadway, the roadway often fails in the weak rock mass. At this time, the weak rock mass is strengthened. Support and weaken the risk of uncoordinated deformation and failure due to different lithological rock masses to enhance the overall stability of the surrounding rock of the roadway.

(6) Starting from the stress field of the surrounding rock, the stress state of the rock mass under the actual triaxial stress state is introduced, and the clarification sequence is reasonable and smooth, but the literature in the past five years accounts for a small proportion of the references, so it is recommended to update further.

Thank you for your advice. In response to your suggestion to add references from the last five years, we have made changes in the revised manuscript and marked the changes with a yellow background.

Replies to reviewer #2:

Authors have conducted experimental study on mechanical response and crack evolution law of coal and sandstone under different stress environments. Before the manuscript can be recommended to be accepted, authors have to make some major revisions as follows: 

(1) Language and grammar errors: L39 “presents the complex stress characteristics”, this sentence lacks a subject. L77 “…for acoustic emission Chen…”

Thank you for your advice. In response to your question, we have asked native English speakers to re-edit the full text of the sentence and mark it with a yellow background in the revised manuscript.

(2) In section Experimental scheme: a) More information about research objects should be provided such as sampling location of rock, basic properties and so on. b) Why the default value of the maximum principal stress σ1 is 17MPa and the minimum principal stress σ3 is 2MPa? How are the four different values of intermediate principal stress selected? It is suggested supply experiment design basis in this section.

Thank you for your advice. In response to your question, we have supplemented the protocol section of the article and marked it with a yellow background. The specific changes are as follows:

The coal and sandstone required for this test come from the Guqiao Coal Mine of Anhui Huainan Mining Group Co., Ltd. The large coal and sandstone body taken on site are processed into a cube shape of 100mm×100mm×100mm (unevenness ≤0.02mm, size error ≤0.2mm) by laboratory means to ensure that the accuracy of the sample meets the requirements, the specific coal and sandstone samples are shown in Fig.3, and the uniaxial compressive strength of the test coal is 21.665MPa, The Poisson's ratio is 0.155, the uniaxial compressive strength of sandstone is 42.608MPa, and the Poisson's ratio is 0.059. The triaxial servo-controlled testing machine preloads the specimen in three directions simultaneously. The preset pressure values in the three directions are reached simultaneously. Then, σ3 and σ2 remain unchanged, and σ1 increases at 30kN/min until the sample is damaged and unstable. According to the schematic diagram of the stress distribution of the surrounding rock of the roadway (Fig.4), it can be seen that due to the change of the principal stress in the process of excavation of the roadway, as the rock mass is far away from the free side, the horizontal stress shows a transition from the direction of stress increase - stress reduction - original stress, at this time, the maximum principal stress is the vertical principal stress (considering the buried depth is 680m, the vertical principal stress σ1=17MPa), the position of the unit body is selected in the roadway gang and near the free side, so the minimum principal stress σ3=2MPa is set, and at the same time, The rock mass at the roadway excavation face has no supporting effect, which forces the rock mass in front of it to produce slow deformation behavior, resulting in changes in the horizontal stress at different positions of the rock mass in front of the roadway (intermediate principal stress σ2). Therefore, to study the influence of different intermediate principal stresses on the failure and deterioration mechanism of the surrounding rock mass, the intermediate principal stresses σ2=3MPa, 6MPa, 10MPa, and 14MPa are set.

Fig.4 Schematic diagram of stress distribution in surrounding rock of roadway

(3) In L211, it would be good to supply references to prove this research results are in good agreement with the existing research.

Thank you for your advice. In response to your question, we have reviewed the relevant references and introduced them into the text, marked with a yellow background.

(4) Conclusions: it is suggested to add one more paragraph introducing the methodology briefly employed in this work, before drawing specific conclusions.

Thank you for your advice. In response to your question, we have added it to the article and marked it with a yellow background. The specific changes are as follows:

In this paper, the theory of deviator stress is introduced to analyze the influence of deviator stress on the plastic zone of the roadway. A triaxial testing machine studies the mechanical properties and acoustic emission evolution law of coal and sandstone with different intermediate principal stresses. The standard and different characteristics of different lithologies under the same stress are analyzed, and the following conclusions are obtained:

---

## [Decision Letter · Decision Letter 1]

22 Oct 2024

Experimental study on mechanical response and crack evolution law of coal and sandstone under different stress environments

PONE-D-24-40527R1

Dear Dr. Xu,

We’re pleased to inform you that your manuscript has been judged scientifically suitable for publication and will be formally accepted for publication once it meets all outstanding technical requirements.

Kind regards,

Xianggang Cheng, Ph.D.

Academic Editor

PLOS ONE

Additional Editor Comments (optional):

Reviewers' comments:

Reviewer's Responses to Questions

**Comments to the Author**

1. If the authors have adequately addressed your comments raised in a previous round of review and you feel that this manuscript is now acceptable for publication, you may indicate that here to bypass the “Comments to the Author” section, enter your conflict of interest statement in the “Confidential to Editor” section, and submit your "Accept" recommendation.

Reviewer #1: All comments have been addressed

Reviewer #2: All comments have been addressed

2. Is the manuscript technically sound, and do the data support the conclusions?

Reviewer #1: (No Response)

Reviewer #2: Yes

3. Has the statistical analysis been performed appropriately and rigorously? 

Reviewer #1: (No Response)

Reviewer #2: Yes

4. Have the authors made all data underlying the findings in their manuscript fully available?

Reviewer #1: (No Response)

Reviewer #2: Yes

5. Is the manuscript presented in an intelligible fashion and written in standard English?

Reviewer #1: (No Response)

Reviewer #2: Yes

6. Review Comments to the Author

Reviewer #1: The authors have revised this paper carefully. I agree with publication.

The authors have revised this paper carefully. I agree with publication.

Reviewer #2: (No Response)

7. PLOS authors have the option to publish the peer review history of their article (what does this mean?). If published, this will include your full peer review and any attached files.

Reviewer #1: No

Reviewer #2: No

---

## [Editor Report · Acceptance letter]

30 Oct 2024

PONE-D-24-40527R1 

PLOS ONE

Dear Dr. Xu, 

I'm pleased to inform you that your manuscript has been deemed suitable for publication in PLOS ONE. Congratulations! Your manuscript is now being handed over to our production team.

Kind regards, 

on behalf of

Dr. Xianggang Cheng 

Academic Editor

PLOS ONE